# Patients with knee osteoarthritis have altered gait and gaze patterns compared to age-matched controls: A pilot study

**Scott Le Rossignol**[1◉]*, **Ewen Fraser**[1‡], **Andrea Grant**[1◉], **Kenji Doma**[2◉], **Matthew Wilkinson**[1‡], **Levi Morse**[1‡], **Peter McEwen**[1‡], **Kaushik Hazratwala**[1‡], **Jonathan Connor**[2◉]

1 Orthopaedic Research Institute of Queensland, Townsville, Queensland, Australia, 2 School of Exercise Science, James Cook University, Townsville, Queensland, Australia

◉ These authors contributed equally to this work.
‡ EF, MW, LM, PM, and KH also contributed equally to this work.
* ortho.pho@oriql.com.au

**Data Availability Statement:** Data for this study cannot be shared publicly because of confidentiality concerns. Data are available from the Orthopaedic Research Institute of Queensland

## Abstract

### Purpose

Although knee Osteoarthritis (KOA) sufferers are at an increased risk of falls, possibly due to impaired gait function, the associated gaze behaviour in patients with KOA are largely unknown. Thus, we compared gait and gaze behaviours characteristics between KOA patients and asymptomatic age-matched controls.

### Results

For Timed Up and Go (TUG) and stair climb tasks, the KOA group demonstrated longer periods of gaze fixations with less frequency of fixations compared to the control group. Conversely, for the Timed up and Go Agility (TUGA) test shorter fixation and frequency patterns were observed. The KOA group presented a shorter final stride length prior to the initiation of the first step in the Stair climb assessment. In addition, for the 30m walk and dual task assessments, the average step length was significantly shorter in the KOA group compared to controls.

### Conclusion

Overall, we found altered gait and gaze behaviours are evident in KOA patients which could relate to their increased falls risk.

## Introduction

Knee osteoarthritis (KOA) is a degenerative joint disease caused by degradation of the cartilage within the knee joint. Incidence and prevalence can be two to tenfold greater in the elderly and is the fourth leading cause of disability worldwide [1, 2]. Patients with KOA have poorer

Ethics Committee (Mater Health Services North Queensland) via email (oriqlresearch@gmail.com) for researchers who meet the criteria for access to confidential data.

**Funding:** The author(s) received no specific funding for this work.

**Competing interests:** The authors have declared that no competing interests exist.

balance capabilities when compared to sex and age-matched counterparts asymptomatic for KOA [3, 4]. Pain, stiffness and reduced knee range of motion are common physically debilitating symptoms which attenuate patient's ability to perform normal gait and activities of daily living and increases the risk of falls. In fact, epidemiological studies demonstrate the risk of falls is greater for individuals with symptomatic KOA compared to healthy older adults [5, 6], particularly during walking [7]. Patients undergoing total knee replacement are at five times increased risk of falls compared to other musculoskeletal disorders [8, 9] with a reported incidence of 40% [10]. Furthermore, falls are directly related to serious morbidity and mortality [11], and place a substantial financial burden on the healthcare system [11]. Understanding balance capabilities of patients with KOA may assist in developing interventions to minimize the risk of falls.

Integration of the sensory and motor pathways within the central nervous system is critical in avoiding falls. The central nervous system has been suggested to play a critical role in the development of KOA [12]. Indeed, KOA patients appear to have impairment in the ability to mentally rehearse gait motor patterns as displayed by fMRI hypoactivation in motor planning (premotor and parietal) brain regions, the brainstem and the cerebellum [13]. Another study demonstrated distinct motor cortex activation by fMRI in KOA patients compared to controls during visually guided force matching motor tasks, with indications of an anterior shift of the knee representation [14]. Alteration in central nervous system functioning in patients with KOA may a pivotal role in balance and falls. Balance capabilities are particularly challenged during complex locomotion, and elderly with KOA exhibit distinct gait characteristics when compared to their asymptomatic counterparts. They particularly have slower gait velocity with shorter steps, longer cycle and longer stance times when ascending stairs [15]. Additionally, they have poorer absorption capabilities in response to forward momentum during a recovery step, with greater hip abduction of lower limbs when stepping over obstacles as compensatory movements [16, 17]. These findings demonstrate that KOA impairs gait functionality and balance.

Dual task walking activities are associated with recurrent falls amongst elderly who live independently [18]. Elderly individuals have impaired dual task performance, even without a severe cognitive impairment, which exacerbates the risk of falling [19]. The risks of falls may be further compounded for elderly patients with KOA, given that they have greater functional impairment at the hip and knee during dual task conditions [16]. The compromise in dual task performance may be underpinned by altered gait and gaze characteristics, although research is yet to confirm this phenomenon.

Balance is a multisensory system which is primarily determined by the interaction between vestibular and visual inputs [20]. Indeed, older adults with high self reported anxiety fixate on targets earlier and longer compared to those with low anxiety [21]. Amongst patients with KOA, fear of falling is associated with sensations of knee instability [22]. Patients with KOA who have an increased fear of falling may also display earlier and longer fixation on targets.

These altered gaze strategies in KOA patients may be explained by impairment in the motor cortex. Furthermore, with recent advances in mobile eye tracking technology, it is surprising that gaze behaviours have not been assessed in patients with KOA as far as we are aware.Gaze behaviours of adults demonstrated that older adults compared to younger adults take longer to initiate steps and have shorter eye fixations during horizontal perturbations designed to off-balance the participant [23]. Impaired proprioception has been demonstrated in patients with KOA which may predispose them to more frequent stepping errors [24]. A comparison of young adults, older adults with low risk of falling and older adults at high risk of falling, showed the reduction in foot placement accuracy was associated with an increased likelihood of falls [25]. Furthermore, the time latency between a saccade onto the target and

toe off was delayed for the high-risk older group when compared to the young adult and low-risk older groups. Additionally, low- and high- risk older groups had a longer duration of fixation on targets compared to the young adult group [26]. Moreover, the older adult group with higher risk of falling have been demonstrated to fixate on more proximal walkways whilst completing an adaptive locomotion task compared to the young adult group [26]. Therefore, older adults at high risk of falling display maladaptive gaze behaviours compared to young adults across a variety of tasks. Given that patients with KOA are at an increased risks of falls, it may be that these patients also display maladaptive gaze patterns. There is currently no research to date investigating gaze patterns amongst patients with KOA. Hence, our aim is to investigate both gait and gaze behaviours of patients with KOA compared to sex and aged matched adults who were asymptomatic of KOA.

## Materials and methods

The study was approved by a recognised Independent Ethics Committee and all participants provided informed written consent (MHS20190113-1). Participants were included into one of two groups if they:

1. had radiographical evidence of severe KOA (as per the Kellgren and Lawrence system) and were waiting to undergo primary total knee replacement (KOA group);

2. were asymptomatic of knee KOA, age and sex matched to recruited KOA group (Age control);

The KOA group were informed that study participation will not affect their management or their timeline to surgery. Participants for each group were excluded if they had a history of;

- vestibular disease

- evidence of dementia

- not capable of undergoing a 2 hour assessment

- unable to stand for greater than 30 minutes or walk without gait aids/assistance

- abnormal central or peripheral vision or lens replacement surgery.

This study is of a cross-sectional design where a control group is compared to the KOA group across several variables. Participants completed a battery of anthropomorphic assessments and maximum voluntary contraction (MVC) with hand held dynamometer. Following these assessments, participants completed both functional and cognitive tasks wearing the portable eye tracking device.

### Maximum voluntary contraction and balance test

The isometric hip and knee strength were maximum voluntary contraction (MVC) force were measured using a handheld dynamometer (HHD) (MicroFET®2 Digital Handheld Dynamometer) with stabilising strap. The HHD was secured in place with a generic velcro body strap and a further strap tethered the assessment leg to a static immovable object (see S1 Fig). Once the examiner made sure the binding was secure, they placed their hand on the HHD without exerting any opposing force and ensuring the belt was kept in place.

During the testing session, the participants underwent a familiarity period to reduce the effect of learning. Knee flexion (KF) and extension (KE) MVC were assessed with participants in a chair sitting upright with their leg placed at 90˚ and feet off the ground. The HHD was

placed on the superior portion of the calcaneal tendon for knee flexion (see S1 Fig) and distal tibia above the ankle joint line for knee extension. Placement of HHD for hip extension (HE) was over the popliteal fossa in a prone position (on a flatbed) and for hip flexion (HF) was over the distal femur proximal to the patella in a seated position. The patients completed three trials for each MVC test and were instructed to perform the test at maximal effort over a period of three seconds. The largest muscular force output recorded out of the three trials were then used for analyses [27]. The Functional Reach task (FRT) [28] was then completed, which involves measuring the difference arm length to maximal forward reach whilst standing directly upright on two feet.

## Gaze analysis

The participants were fitted with a portable eye tracker (Mobile eye XG, Applied Science Laboratories, Massachusetts, USA) for the functional and cognitive tasks. Calibration was accomplished following manufacturer guidelines. Analysis was conducted using the built-in eye tracking software (ASL results plus; applied science laboratories, Bedford MA). The number of fixations and duration of fixations was calculated from the software where the minimum time for fixation was 100ms. The gaze tracking system has shown acceptable discriminant validity [29], and has an accuracy of up to 0.5 degrees of visual angle.

## Gait analysis

The participants' gait was recorded by an Apple iPad in sagittal plane at 220 Hz, which was operated by one investigator to ensure generalizability to the clinical setting. The iPad was placed in a position to ensure the smallest possible frame of view but still capture the entirety of the locomotor tasks. However, for the 30m walk and dual tasks, the iPad was positioned to capture 2–3 steps in the middle portion of the tasks. Kinovea 0.8.25 (open-source motion analysis software), which has been demonstrated as being reliable and valid up to 5m from the object [30] was utilized to measure the step length and swing duration. Step length was measured at time of heel strike between both heels and swing duration was measured as time of toe off to time of heel strike. For the stair task,the distance was also measured between the most anterior toe of the final step prior and the base of the stairs.

## Locomotor tasks

Locomotor tasks included a Stair climb task, Timed Up and Go (TUG) and Timed Up and Go Agility (TUGA) test. Participants were instructed to perform tasks as quickly as possible for three trials of each task. The stair climb task involved walking 3m towards a custom built step apparatus, walking up three steps each measuring 18 x 18cm pivoting on the top step before descending to finish at the starting position. In addition to the standardized TUG [31] we included a modified time up and go test with an agility component (TUGA), which added three cones in a straight line 1m apart. The participant completes this test the same as the TUG test however the participant weaved between the cones both up and back to the starting position.

## Cognitive dual tasks

There were three cognitive dual tasks which involved participants completing the dual activity of walking a 30m course whilst [1] carrying 180ml of water in a 200ml cup volume (cup with water), [2] counting down numbers in sequence from 100 in 7's (serial numbers) and [3] naming different animal species (serial animals). Participants were instructed to complete each dual walking task as quickly as possible. The 30m walk course was a 15m bi-directional track.

**Table 1. Demographics of participants.**

|  | Control N = 10 | KOA N = 11 | P value |
|---|---|---|---|
| Age (+/- SD) | 72 (5) | 68 (8) | t score 1.27 (0.22) |
| Sex (M/F) | 3/7 | 3/8 | Chi square 0.97 (0.3) |
| Height (+/- SD) | 154 (33) | 163 (8) | t score 0.73 (0.47) |
| Weight (+/- SD) | 91 (32) | 87 (17) | t score 0.47 (0.64) |

## Statistical analysis

The measure of central tendency and dispersion for all parameters was reported as median and inter-quartile range, and the data was analysed using the Statistical Package of Social Sciences (v25, IBM, Chicago, USA). All parameters were compared between groups using the Mann-Whitney U test, given that we conducted a pilot study with a small sample size [32]. The the alpha level set at 0.05. The magnitude of differences between KOA and Control groups were also determined using effect size calculations (Cohen's d), with values of 0.2, 0.5 and 0.8 considered as small, moderate and large effect size (ES) calculations [33].

## Results

Twenty one participants [15 female, 6 male with age mean = 69.5 and SD = 6.9) were identified from investigating orthopaedic surgeons private practices (three sites) and from local community institutions via convenience sampling (Table 1). There were 10 older adults asymptomatic of KOA allocated into control group (age 72 ± 5; height 154 ± 33; body mass 91 ± 32) and 11 diagnosed with severe KOA were allocated into the KOA group (age 68 ± 8; height 163 ± 8; body mass 87 ± 17)).

### Time-to-completion of locomotor and cognitive tasks, MVC and functional reach

There was no differences between the groups for height, weight, age and sex. The time-to-completion tasks were achieved significantly faster for the locomotor activities in the control group for 30-m walk, serial animal, stair climb, TUG and TUGA, with moderate to large ES (Table 2). However, no significant differences were found between KOA and control groups in the cup with water and serial numbers tasks, although ES calculations were moderate.

The knee extension MVC was significantly greater for the control group than the KOA group for the both lower limbs with large ES (Table 3). There were no between-group differences in knee flexion MVC for the right limb, although ES was moderate on the right and large on the left and there was trend towards significance on the left side. Furthermore, no between-group differences were identified for hip flexion MVC, hip extension MVC and the functional

**Table 2. Time-to-complete: KOA vs Control.**

|  | KOA (median [IQR]) | Control (median [IQR]) | ES | P-value |
|---|---|---|---|---|
| Walk 30m (sec) | 28.6 (27.9–31.7) | 23.0 (21.3–26.5) | 1.69 | **0.003** |
| Cup with water (sec) | 37.3 (33.5–42.4) | 33.4 (28.2–38.6) | 0.54 | 0.17 |
| Serial numbers (sec) | 32.0 (29.5–43.5) | 27.2 (25.9–33.9) | 0.54 | 0.11 |
| Serial animal (sec) | 31.1 (28.6–35.2) | 26.4 (24.6–29.8) | 0.59 | 0.05 |
| Stair climb | 11.4 (8.8–15.1) | 7.4 (7.2–8.9) | 1.70 | **0.001** |
| TUG | 9.8 (9.2–10.8) | 6.6 (6.0–7.4) | 2.32 | **<0.001** |
| TUGA | 12.0 (10.3–12.9) | 8.1 (7.3–9.3) | 2.11 | **0.001** |

**Table 3. MVC scores of hips and knees and functional reach test scors.**

| Task | | KOA (median [IQR]) | Control (median [IQR]) | ES | P-value |
|---|---|---|---|---|---|
| KF MVC | Right | 6.5 (4.7–13.0) | 12.8 (4.8–19.6) | 0.52 | 0.25 |
| | Left | 6.4 (3.0–12.4) | 14.0 (4.6–19.0) | 0.96 | 0.05 |
| KE MVC | Right | 11.3 (10.3–28.8) | 36.1 (19.4–40.0) | 1.14 | **0.02** |
| | Left | 13.3 (11.8–26.6) | 31.7 (21.2–38.6) | 0.93 | **0.04** |
| HF MVC | Right | 23.3 (7.9–28.4) | 31.1 (15.0–36.5) | 0.66 | 0.07 |
| | Left | 19.6 (9.3–29.6) | 27.0 (17.0–29.2) | 0.60 | 0.43 |
| HE MVC | Right | 17.8 (10.9–25.2) | 22.9 (15.9–32.7) | 0.40 | 0.32 |
| | Left | 17.1 (8.2–21.6) | 21.5 (13.0–33.3) | 0.61 | 0.25 |
| FRT | Right | 32.3 (30.5–36.5) | 37.4 (30.7–39.8) | 0.38 | 0.32 |
| | Left | 31.0 (25.1–34.5) | 34.0 (28.6–35.8) | 0.56 | 0.32 |

reach test, although ES calculations were moderate-to-large. However, hip extension MVC for the right limb and the functional reach test for the left and right limbs exhibited small ES. Furthermore, there was no difference between groups for FRT, with a small effect on the right side and moderate on the left side.

### Gaze metrics during locomotor tasks

We were unable to analyse gaze data from two participants due to technological issues with the eye tracking hardware. The KOA group demonstrated significantly more fixations, and shorter fixation durations than the control group during the TUGA task (Table 4). Conversely, the KOA group demonstrated significantly less frequent but longer fixation durations, than the control group during the TUG and stair climbing tasks. ES calculations between KOA and control group indicated moderate to large ES for each locomotor activity.

### Gaze metrics during cognitive dual tasks

There were no differences for fixation frequency or duration during the cognitive dual tasking activities (Table 5). ES calculations were also small between group means, except a moderate effect size identified for fixation duration during serial animals.

### Gait metrics during stair climb task

During stair climb task with the stairs, participants with KOA had a significantly shorter final step length before stepping up onto the stairs (Table 6). There were no other significant differences for step length or swing time. ES was small for average swing duration, moderate for average step length and large for final step length onto stairs.

**Table 4. Number of fixations per second and fixation duration during the locomotor tasks.**

| Locomotor Task | Gaze Metric | KOA (median [IQR]) | Control (median [IQR]) | ES | P value |
|---|---|---|---|---|---|
| TUG | Fixations | 4.8 (4.5–5.5) | 5.8 (3.0–6.7) | 1.11 | **0.02** |
| | Durations | 0.20 (0.18–0.25) | 0.17 (0.15–0.33) | 1.30 | |
| TUGA | Fixations | 6.6 (6.3–7.5) | 5.8 (5.4–6.4) | 0.97 | **0.03** |
| | Durations | 0.15 (0.13–0.16) | 0.17 (0.16–0.18) | 0.95 | |
| Stair climbing | Fixations | 4.6 (4.1–5.5) | 6.2 (4.6–7.4) | 0.91 | **0.04** |
| | Durations | 0.22 (0.18–0.24) | 0.16 (0.14–0.22) | 0.73 | |

**Table 5. Number of fixations per second and fixation duration during the dual cognitive tasks.**

| Dual Tasks | Gaze Metric | KOA (median [IQR]) | Control (median [IQR]) | ES | P value |
|---|---|---|---|---|---|
| Control 30m Walk | Fixations | 4.8 (4.4–6.5) | 5.3 (4.5–6.8) | 0.36 | 0.53 |
| | Durations | 0.20 (0.15–0.23) | 0.19 (0.15–0.22) | 0.39 | 0.53 |
| Serial numbers | Fixations | 4.2 (2.8–6.1) | 5.2 (4.3–6.5) | 0.47 | 0.35 |
| | Durations | 0.24 (0.17–0.36) | 0.19 (0.15–0.24) | 0.5 | 0.39 |
| Serial Animals | Fixations | 6.8 (3.0–8.1) | 5.2 (5.2–5.9) | 0.21 | 0.53 |
| | Durations | 0.15 (0.12–0.33) | 0.19 (0.17–0.19) | 0.11 | 0.53 |
| Cup with water | Fixations | 2.2 (1.1–5.3) | 4.2 (1.9–5.5) | 0.47 | 0.53 |
| | Durations | 0.46 (0.19–0.89) | 0.24 (0.18–0.54) | 0.08 | 0.53 |

**Table 6. Average step length, average swing duration and final step length from stairs within the stair climb tasks between the KOA and control groups.**

| Gait behaviour | KOA (median [IQR]) | Control (median [IQR]) | ES | P value |
|---|---|---|---|---|
| Average Step length | 52.46 (47.75, 58.09) | 56.11 (52.65, 63.36) | 0.70 | 0.12 |
| Average swing duration | 0.38 (0.34, 0.40) | 0. 36 (0.34, 0.38) | 0.40 | 0.40 |
| Final step length from stairs | 21.04 (16.25, 26.14) | 28.89 (21.95, 38.10) | 1.13 | **0.03** |

**Table 7. Comparisons of gait metrics for dual cognitive tasks between the KOA and control groups.**

| Locomotor Task | Gait behaviour | KOA (median [IQR]) | Control (median [IQR]) | ES | P value |
|---|---|---|---|---|---|
| 30m Walk | Average Step length | 57.62 (54.59, 61.90) | 71.58 (69.92, 76.46) | 2.09 | **0.003** |
| | Average Swing duration | 0.37 (0.33, 0.39) | 0.36 (0.33, 0.38) | 0.25 | 0.374 |
| Serial Animals | Average Step length | 56.87 (53.74, 60.38) | 66.37 (62.66, 74.27) | 1.62 | **0.005** |
| | Average Swing duration | 0.38 (0.36, 0.39) | 0.36 (0.34, 0.37) | 0.77 | 0.149 |
| Serial Numbers | Average Step length | 55.82 (51.52, 57.72) | 65.52 (61.34, 69.60) | 1.90 | **0.002** |
| | Average Swing duration | 0.41 (0.36, 0.41) | 0.36 (0.35, 0.37) | 1.04 | 0.041 |
| Cup with water | Average Step length | 52.57 (50.08, 54.66) | 56.79 (51.66, 63.56) | 1.22 | 0.051 |
| | Average Swing duration | 0.37 (0.36, 0.39) | 0.36 (0.34, 0.40) | 0.36 | 0.380 |

## Gait metrics during dual cognitive tasks

The KOA group had significantly shorter step lengths (Table 7) compared to the control group during the 30m walking task and cognitive dual tasks (serial numbers and serial animal). The average swing duration for the serial numbers task was significantly shorter for the KOA group compared to the control group. No other significant differences were reported.

## Discussion

The aim of this study was to compare the gaze and gait metrics of KOA patients with an age-matched, otherwise healthy group. This has been achieved by demonstrating moderate to large differences in effect size for gait and gaze behaviour between KOA and healthy age-matched controls during various task-specific scenarios. Time-to-completion was less with large differences in effect size for the control group for the TUG, TUGA, stair climb and 30m walk. However, only moderate differences in effect size was identified in time-to-completion during the serial numbers, serial animal and cup with water tasks between groups. The control group had greater knee extension MVC for both left and right limbs and greater knee flexion MVC for the left limb with large effect size. However, there were only small to moderate effect sizes for

right knee flexion, bilateral hip flexion and extension. Participants with KOA had a shorter final step onto stairs than controls with a large effect size. They also had shorter average step length than controls in the 30m walk, serial animals and serial numbers tasks, all with large effect sizes. The KOA group had a longer average swing duration for the serial numbers tasks than controls with a large effect size. Additionally, participants with KOA demonstrated a greater number of fixations, occurring for a shorter duration, than healthy age-matched control participants in the TUG, agility TUG, and staircase task, all with large effect size. However, the differences between groups during cognitive dual tasks and the 30m walk for number of fixations or duration of fixations exhibited small effect size. Together, these findings highlight that orthopaedic dysfunctions perturbs gait function, which may be underpinned by altered gaze behaviour during locomotor activities.

With respect to gaze behaviour, the KOA group had less fixations and longer duration of fixations during TUG and stair climb tasks and more fixations and shorter duration of fixations in the TUGA task compared to the control group, all with large effect size. It is interesting that the gaze patterns of the TUGA task was reversed to that of the TUG and stair climbing tasks. This may be related to the nature of the TUGA task involving weaving around cones and thereby more targets to fixate on leading to more fixations for shorter duration. Conversely, for the TUG and stair climbing tasks, the participants were able to constrain their gaze to a greater degree. Due to the novelty of this pilot study synthesis with current research is limited. Zukowski et al. (2020) compared fallers and non-fallers, showed that fallers had shorter duration of fixation on immediate surroundings and that they walked slower than non-fallers [34]. They suggested that the fallers possibly prioritize the near environment ahead of the far which is possible as a result of increased anxiety. Shorter duration of fixation is displayed by participants identified as fallers and participants with KOA during the TUGA task thus suggesting the KOA may be a subgroup of the heterogenous fallers. Previous work has highlighted that the gaze behaviour of individuals with dysfunctional locomotion (i.e., older adults at higher risk of falls) often demonstrate premature fixation away from the target they are stepping towards, in order to fixate on a proximal location in their walking trajectory [26, 35]. These gaze strategies are thought to represent a prioritization of visual information used to consciously control each locomotor action [26]. These findings indicate individuals with KOA may either possess or develop suboptimal gaze strategies when engaged in locomotor activities.

The participants with KOA had longer time-to-completion for the 30m walk, serial animals, stair climb, TUG and TUGA compared to controls, all with large effect size. This suggests the KOA group had a reduced functional capacity compared to controls which is possibly a result of increased pain during locomotion over a number of years leading to increasing levels of deconditioning. There was no difference in time-to-completion for the cup with water and serial numbers task. It seems the most plausible explanation is that these tasks impaired the walking velocity of each group. However, the control group had a shorter 30m walk time-to-completion and therefore the impairment on gait velocity in the control group was greater than that of the KOA group. Moreover, the KOA group had weaker knee extension MVC bilaterally compared to the control group. Quadricep strength and therefore knee extension MVC when low, is considered a risk factor for osteoarthritis independent of pain [36]. There was no difference of knee flexion, hip extension and hip flexion MVC bilaterally between groups.

Regarding effect size between groups for the FRT, on the right it was small and left it was moderate. Our results are consistent with previous findings showing that the FRT may not be a valid indicator of dynamic balance during gait by showing a poor correlation with free gait speed and no difference between healthy controls and those with a vestibular impairment [37].

Despite this, the FRT has demonstrated an ability to differentiate between patients with Parkinson's with and without a fall's history [38] and concluded that it is a useful test for identifying those at an increased risk of falls as part of a battery of balance tests [38]. Previous work has demonstrated that KOA patients have greater postural sway, balance symmetry and mean velocity centre of pressure on balance platforms [39, 40]. The intention of using the FRT in this study was to utilize an assessment that was accessible and cost-effective, an appropriate testing procedure in the clinical setting. However, the FRT may not be appropriate as a standalone protocol to determine risks of falls for patients with severe KOA [41].

The KOA group demonstrated no difference in average step length or step duration during the tasks involving stairs, which was likely related to inclusion of the acceleratory phase of walking as a result of starting from stationary. This is inconsistent with previous findings [42], whereby females with patellofemoral OA have shorter average step length, however their protocol excluded acceleratory and decelerator phases and are therefore not comparable. The KOA group demonstrated significantly shorter final step onto the stairs than the control group. We believe this is consistent with Stefanik et el. (2016) whereby patients with KOA have a reduced maximal step length. The KOA group had shorter average step length during the 30m walk, naming animals and serial numbers tasks, all with large effect size, which is consistent with previous research [42]. During these tasks, the video recording was in the middle of the task so the participant had likely reached a "steady state" velocity. There was a trend towards a shorter average step length for the KOA group during the cup with water dual task (p = 0.051), with a large ES. Moreover, the difference of average step length between the 30m walk and cup with water dual task was approximately 6cm for the KOA group and 23cm for the control group. This suggests that both groups reduced their step length in order to prevent spilling of water from the cup, where the reduction of step length was greater in the control group because they had a longer average step length during the 30m walk.

During the serial numbers task, the swing duration was longer for the KOA group in comparison to the control group with a large effect size, although there was no difference in swing duration during the 30m walk, naming animals or cup with water tasks between the KOA and control groups. The difference in swing duration during the serial numbers tasks could possibly be explained by perceived difficulty of this task compared to the other dual tasks, which may have impeded walking velocity to a greater degree amongst the KOA group compared to the control group. Indeed, the average swing duration is similar amongst all dual tasks in the KOA group but there was less variation within the serial numbers task, possibly suggesting that amongst the KOA participants the impairment of cognitive difficulty was more uniform during this task. This is partially consistent with previous research [43], that showed angular velocity positively correlating with KOOS, hence participants with worse knee symptomology had slower angular velocity and therefore slower swing duration.

When analyzing gaze under increased cognitive load, no difference in gaze behaviour metrics was found between groups. Ellmers et al., (2016) previously analyzed gaze behaviour whilst simultaneously carrying out the secondary serial subtraction task, and reported slower movement completion rates and suboptimal gaze strategies [44]. Furthermore, maladaptive gaze strategies observed in high-risk older adults (e.g., individuals with KOA) may also be due to limited working memory or attentional processing deficiencies [35]. There is a need for future work to explore whether individuals with severe KOA have impairment of working memory and attentional processing capacity, resulting in greater gait and gaze behaviour deficiency.

Shorter average step length across dual tasks and shorter final step prior to stairs suggests that participants with KOA overall have a shorter step length during locomotion. This may be restrictive of these participants from being able to execute adaptive maneuverers in order to avoid obstacles thus increasing their risk of falls and also their fear of falling, i.e the control

group may have the ability to either shorten or lengthen their next step in order to avoid obstacle whereas the KOA group may only be able to reduce the length of their next step which is already shorter than that of the control group. This reduced number of options may have an increased cognitive load to process and plan future movements. It is not clear why the dual cognitive tasks appeared to impede the step length of participants to a greater extent than that of the control group. Perhaps they have reduced proprioceptive feedback and therefore rely greater on cognitive processing to continue locomotion. Alternatively, pain whilst walking may interrupt cognitive processing leading to instability and increased risk of falls. in the elderly [45].

From the clinical perspective, knee instability is a common symptom of KOA which has been associated with both falling and fear of falling [22, 46]. Maladaptive gaze behaviours described above may be the result of the increased fear of falling in KOA through limbic mediated regulation. Alternatively, dysfunctional proprioception present in KOA may lead to adaptation of neural pathways dysregulation of motor nerves to extra-ocular muscles. Further elucidation of this mechanism may assist with rehabilitation or prevention of falls amongst patients with KOA. Potential targets for this may include rehabilitation focusing on proprioception, focus on strength to improve knee stability or alternatively directly targeting gaze behaviours through visual cues. Young and Hollands (2010) showed that an intervention to delay gaze transfer was successful in reducing stepping errors and suggested that suboptimal visual sampling contributed to falls [45]. They proposed that elderly at high risk of falling look away early from stepping targets in order to fixate future constraints in their walking path. A similar study substituting elderly for patients with KOA could establish a similar relationship. Curzon-Jones and Hollands (2018) showed that previewing a route with obstacles improved self confidence but had no differences of anxiety and that this previewing led to a reduction of stepping errors [47]. They suggested that older adults prioritise future stepping constraints over ongoing stepping actions. KOA patients are also likely to display this maladaptive prioritisation present in older adults and may be even further exaggerated. Chandra et al. (2011) suggested that elderly required a longer period of gaze on obstacles in order to gain sufficient visual information possibly due to deterioration of peripheral visual acuity [35]. This is partly consistent with our findings, KOA participants have both a greater number of fixations and shorter duration of fixations, likely means that the total duration of fixations is not different from age matched controls. Given, that the KOA participants likely have the same peripheral visual acuity as elderly, suggests that both groups are comparable in terms of required time of visual sampling. Chapman and Hollands (2006) showed that elderly at high risk of falling had a longer duration of fixation and suggested that fixation duration may be inversely related to walking speed [26].

Clinically, analysis of gait may help to identify KOA patients at an increased risk of falling. Targeting muscle strength may help to increase the step length of those most at risk of falls thereby reducing their risk of falls, indeed improved leg muscle strength has been shown to reduce the risk of falls [48]. Education of KOA patients to be wary of completing difficult cognitive tasks whilst walking may also reduce the risk of falls. Lin and Lin (2016) have shown reduced step length in young participants whilst performing working memory tasks [49]. Indeed, this is consistent with our findings however it seems that the KOA group had a greater reduction in step length. This suggests that cognitive tasks may impede the KOA group to a greater extent than the age-matched controls. This may be due to more afferent information is required in the KOA participants because of the dependence on visual information as a result of decreased proprioceptive information.

Being a pilot study there is a small cohort which is a limitation, despite this we have had large effect sizes in various analysis which will direct future research. Another limitation is that

the control group have not had knee xrays to examine if they have KOA however they were asymptomatic. As we know there is often a mismatch of radiographic evidence of KOA and symptoms. Clinically, pain interfering with activities of daily in conjunction with radiographic evidence of KOA is an indication for a total knee arthroplasty, whereas radiographic features without pain are not indication. Furthermore, we believe that pain is related to fear of falling and central nervous system dysfunction. Despite these limitations, large effect were found across various gait and gaze parameters warrants further investigation in a larger scale study.

## Conclusion

This is the first study to our knowledge assessing both gait and gaze metrics between KOA and age-matched control participants. The results of this study show that KOA patients gaze fixations are prolonged and less frequent than their age-matched controls for TUG and stair climbing tasks. The TUGA task conversely, demonstrated more fixations of shorter duration of fixations in the KOA compared to the control group. In terms of gait metrics, there was a significantly shorter final step onto stairs in the KOA compared to the control group during the stair climbing task. In addition, average step length was significantly shorter in the KOA group during the 30m walk as well as dual tasks serial animal animals and serial numbers compared to controls. Time-to-complete tasks were significantly longer in the KOA group during the 30m walk, serial animals, stair climb, TUG and TUGA tasks. The data suggests that when combining cognitive tasks with functional movement KOA patients show impeded gait to a greater extent than just age associated related changes. Thus, the combination of functional movement with higher cognitive functioning may contribute to an increased risk of falling. Further research is required within the area of gaze and gait metrics amongst patients with KOA in order to identify those at an increased risk of falling or decrease their risk of falling.

## Supporting information

**S1 Fig. Demonstration of isometric maximal force contraction for knee flexion.** Dynamometer is placed at the posterior distal calf with the stabilizing strap.
(DOCX)

## Acknowledgments

Dr Reece Tso–For his contribution to data collection

## Author Contributions

**Conceptualization:** Andrea Grant, Kenji Doma, Matthew Wilkinson, Levi Morse, Peter McEwen, Kaushik Hazratwala, Jonathan Connor.

**Data curation:** Andrea Grant.

**Formal analysis:** Kenji Doma, Jonathan Connor.

**Investigation:** Scott Le Rossignol, Andrea Grant, Kenji Doma, Jonathan Connor.

**Methodology:** Scott Le Rossignol, Ewen Fraser, Andrea Grant, Kenji Doma, Jonathan Connor.

**Supervision:** Andrea Grant, Kenji Doma, Matthew Wilkinson, Levi Morse, Peter McEwen, Kaushik Hazratwala, Jonathan Connor.

**Writing – original draft:** Scott Le Rossignol, Kenji Doma, Jonathan Connor.

**Writing – review & editing:** Scott Le Rossignol, Ewen Fraser, Andrea Grant, Kenji Doma, Jonathan Connor.

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
