## [Decision Letter · Decision Letter 0]

26 Oct 2022

PONE-D-22-25749Patients with Knee Osteoarthritis have altered Gait and Gaze Patterns compared to Age-Matched Controls: A Pilot Study.PLOS ONE

Dear Dr. Le Rossignol, 

Thank you for submitting your manuscript to PLOS ONE. After careful consideration, we feel that it has merit but does not fully meet PLOS ONE’s publication criteria as it currently stands. Therefore, we invite you to submit a revised version of the manuscript that addresses the points raised during the review process.

We look forward to receiving your revised manuscript.

Kind regards,

Ravi Shankar Yerragonda Reddy, Ph.D

Academic Editor

PLOS ONE

Journal Requirements:

3. Please amend the manuscript submission data (via Edit Submission) to include author  Kaushik Hazratwala.

4. We note you have included a table to which you do not refer in the text of your manuscript. Please ensure that you refer to Table 5 in your text; if accepted, production will need this reference to link the reader to the Table.

Reviewers' comments:

Reviewer's Responses to Questions

**Comments to the Author**

1. Is the manuscript technically sound, and do the data support the conclusions?

Reviewer #1: Partly

Reviewer #2: Partly

2. Has the statistical analysis been performed appropriately and rigorously? 

Reviewer #1: No

Reviewer #2: I Don't Know

3. Have the authors made all data underlying the findings in their manuscript fully available?

Reviewer #1: Yes

Reviewer #2: Yes

4. Is the manuscript presented in an intelligible fashion and written in standard English?

Reviewer #1: Yes

Reviewer #2: No

5. Review Comments to the Author

Reviewer #1: I want to thank the authors for their important work. This topic is of great interest, and is close to my heart. I can tell the design, performance, and analysis of the study, along with the preparation of this manuscript, took a lot of time and effort. Thank you for the opportunity to review this manuscript.

Overall, I think it's important to remember the authors present this as a pilot study, which is something I think gets lost when communicating the findings of this current investigation. This is especially true in the way the discussion and conclusions are presented. I would assume this study will be followed up with a larger, appropriately powered investigation, hence the absence of an a priori analysis for this study. If the methods and sample size are designed for a pilot, then the discussion and conclusions should be more aimed at discussing and making conclusions aimed at preparing for the next study, and not on trying to make definitive statements.

I think a quick proofread for grammatical errors is appropriate, (e.g., lines 15 "behavior" should be "behaviors", 20 remove "more", 135 "climba" should be "climb", etc.).

INTRODUCTION

Refs 12, 13, 14 are appropriate given the topic, but there are some newer publications that would also be beneficial to include.

The authors do not state the alternate hypothesis in the INTRODUCTION, and only state it formally in the DISCUSSION (line 219-220).

METHODS

Lines 74-79 which describe descriptive data of the participants is really more appropriate in the RESULTS section.

Controls were "apparently healthy." Can you operationally define this?

Experimental group were those with "severe KOA." What severity grading was used? Kellgren-Lawrence? This needs to be stated and included.

How was age matching performed (e.g., exact age or +/- a certain number of years)? Was sex matching performed as well? The authors state in the results there was no difference in sex overall. We know sex has an effect of KOA. I want to know if you matched a 45 year old male to a 45 year old female, or if they also had to have the same sex.

You state the participants were those who walk "independently," which would indicate none of them required the use of an assistive device (i.e., canes, walkers, etc.). Please clarify.

Lines 113-114: You state that you took the largest value and used for analysis. This was done for other dependent variables as well. I'm more familiar with using the average across three trials. As long as you can provide a reference stating this has been done prior, then I would be fine with this.

Please provide reliability and validity for the system used in the gait analysis.

Why were non-parametric statistical analyses performed? I would assume this was due to not meeting the assumptions for parametric analysis, which could be a result of the small sample size. Regardless, please provide details of the assumptions analysis and why you chose this method.

There are a lot of comparisons performed in this investigation, so setting your alpha at .05 really runs the risk of increasing your family-wise error rate. The authors should not feel pressured to "find significance" in a pilot study. Again, the main purpose will be to provide effect sizes to be used in an a priori analysis for the NEXT study and to make improvements on the methodology going forward. I would rather see you be more conservative (e.g., Bonferroni correction per analysis).

RESULTS

Please provide another table with descriptive statistics of the participants and perform statistical comparisons between the groups (age, height, weight, etc.).

DISCUSSION

The discussion is quite lengthy, but well-written. If the authors are struggling with a word count while trying to address any comments, I would recommend cutting this down a bit.

Please add a Limitations section prior to your Conclusion.

Reframe your Discussion and Conclusions per the pilot nature of this study. This is probably not a definitive investigation on this topic. Temper your conclusions appropriately. See notes above.

Reviewer #2: This submission examines the gaze behavior during ambulation tasks of persons with osteoarthritis of the knee and compares these behaviors (as well as some gait metrics) with age matched controls without OA of the knee. The submission shares some interesting observations but it suffers from the lack of a central theory and in its current state, does not seem to present or test a clear hypothesis.

The introduction simply states that people with OA present with weakness, pain, abnormal gait and are at an increased risk for falls. It follows with a discussion of the presence of differences in gaze behaviors and balance when comparing older adults at risk for falls and younger people and finishes with a proposal that gaze behaviors in persons with knee OA have not been studied. There is no discussion of a proposed relationship between living with knee OA and the development of maladaptive gaze behaviors. In the absence of a literature on such a specific question, presenting a broader discussion of CNS adaptations to chronic orthopedic conditions could help a reader make the connection between the two phenomena being discussed. Without this grounding in theory it is difficult to see this submission as something beyond a report of two co-occurring conditions in older persons.

This submission presents dozens of statistical comparisons. The authors should consider focusing their paper to test a primary hypothesis and identify some key secondary hypotheses and analyses. Sticking with the current shot gun approach would require presenting corrected p-values.

The description of the two samples needs to be more extensive. Critical information such as subjects and controls comorbidities (particularly DM), fall history, and the presence or absence of other orthopedic issues needs to be shared, as does some measure of the severity of knee OA in the OA group and the length of time they have suffered from OA.

More information is needed on the system utilized to collect gaze behaviors. How does the equipment work? Is there published data speaking to the reliability and validity of measurements collected during gait, with the system, in the studied population?

The analyses of dual task data might make more sense if the authors analyzed the differences in dual task cost as opposed to citing abnormal single task data, abnormal dual task data and proposing a possible central neural difference causing the abnormal dual task data.

The FRT analysis does not fit into this paper. A static limit of stability test with the subject in a wide stance does little to inform a discussion of stability in gait, a much higher order of balance. I could make a similar argument against keeping the MVC analyses. People with KOA present with decreased force generation abilities…..

6. PLOS authors have the option to publish the peer review history of their article (what does this mean?). If published, this will include your full peer review and any attached files.

Reviewer #1: No

Reviewer #2: No

---

## [Author Response · Author response to Decision Letter 0]

20 Dec 2022

Reviewers comments and authors responses 

Reviewer’s comment: 

Author’s response: Thank you for sending through the links for formatting style. The author/s have reviewed the formatting templates and modified the manuscripts accordingly. 

Reviewer’s comment: 

Author’s response: The Authors ORCID iD is 0000-0002-8324-6801

Reviewer’s comment: 

3. Please amend the manuscript submission data (via Edit Submission) to include author Kaushik Hazratwala.

Author’s response: The author will be sure to add the above author in the manuscript submission.

Reviewer’s comment: 

4. We note you have included a table to which you do not refer in the text of your manuscript. Please ensure that you refer to Table 5 in your text; if accepted, production will need this reference to link the reader to the Table.

Author’s response: Thank you for notifying the author of this. Table 5 is now referred to within the text. 

 

Reviewer's Responses to Questions

Comments to the Author

1. Is the manuscript technically sound, and do the data support the conclusions?

Reviewer #1: Partly

Reviewer #2: Partly

2. Has the statistical analysis been performed appropriately and rigorously?

Reviewer #1: No

Reviewer #2: I Don't Know

3. Have the authors made all data underlying the findings in their manuscript fully available?

Reviewer #1: Yes

Reviewer #2: Yes

4. Is the manuscript presented in an intelligible fashion and written in standard English?

Reviewer #1: Yes

Reviewer #2: No

5. Review Comments to the Author

Reviewer #1: I want to thank the authors for their important work. This topic is of great interest, and is close to my heart. I can tell the design, performance, and analysis of the study, along with the preparation of this manuscript, took a lot of time and effort. Thank you for the opportunity to review this manuscript.

Reviewer’s comment: 

Overall, I think it's important to remember the authors present this as a pilot study, which is something I think gets lost when communicating the findings of this current investigation. This is especially true in the way the discussion and conclusions are presented. I would assume this study will be followed up with a larger, appropriately powered investigation, hence the absence of an a priori analysis for this study. If the methods and sample size are designed for a pilot, then the discussion and conclusions should be more aimed at discussing and making conclusions aimed at preparing for the next study, and not on trying to make definitive statements.

Author’s response: Thank you for acknowledging the time taken to produce this manuscript. The discussion and conclusion have been reworded to be more in line with being a pilot study. 

Reviewer’s comment: I think a quick proofread for grammatical errors is appropriate, (e.g., lines 15 "behavior" should be "behaviors", 20 remove "more", 135 "climba" should be "climb", etc.).

Author’s response: The manuscript has now been proof read and the above errors corrected.

INTRODUCTION

Refs 12, 13, 14 are appropriate given the topic, but there are some newer publications that would also be beneficial to include.

Reviewer’s comment: The authors do not state the alternate hypothesis in the INTRODUCTION, and only state it formally in the DISCUSSION (line 219-220).

Author’s response: The alternate hypothesis has now been removed from the manuscript as this is a pilot study and therefore no hypothesis is required. 

METHODS

Reviewer’s comment: Lines 74-79 which describe descriptive data of the participants is really more appropriate in the RESULTS section.

Author’s response: The lines above have been moved to the results section. 

Reviewer’s comment: Controls were "apparently healthy." Can you operationally define this? 

Author’s response: “Apparently healthy” has been removed from the manuscript due to being unable to operationally define this term. 

Reviewer’s comment: Experimental group were those with "severe KOA." What severity grading was used? Kellgren-Lawrence? This needs to be stated and included.

Author’s response: Thank you for this suggestion. The Kellgren-Lawrence was utilized and has been stated and included in the manuscript. 

Reviewer’s comment: How was age matching performed (e.g., exact age or +/- a certain number of years)? Was sex matching performed as well? The authors state in the results there was no difference in sex overall. We know sex has an effect of KOA. I want to know if you matched a 45 year old male to a 45 year old female, or if they also had to have the same sex.

Author’s response: Thank you for highlighting this omission. Participants were matched by sex and age which has now been included in the manuscript. 

Reviewer’s comment: You state the participants were those who walk "independently," which would indicate none of them required the use of an assistive device (i.e., canes, walkers, etc.). Please clarify.

Author’s response: Thank you for highlighting this lack of clarity. The manuscript has removed independently and replaced it with; unable to stand for greater than 30 minutes or walk without gait aids/assistance 

Reviewer’s comment: Lines 113-114: You state that you took the largest value and used for analysis. This was done for other dependent variables as well. I'm more familiar with using the average across three trials. As long as you can provide a reference stating this has been done prior, then I would be fine with this.

Author’s response: Maximal voluntary contraction requires the largest value be used for analysis a reference has been supplied within the manuscript. 

Reviewer’s comment: Please provide reliability and validity for the system used in the gait analysis.

Author’s response: The manuscript has been edited to provide adequate reliability and validity. 

Reviewer’s comment: Why were non-parametric statistical analyses performed? I would assume this was due to not meeting the assumptions for parametric analysis, which could be a result of the small sample size. Regardless, please provide details of the assumptions analysis and why you chose this method.

Author’s response: With respect to the statistical analyses, we used non-parametric analyses as we conducted a pilot study with a small sample size (Altman et al., 1983; PMID: 6405856). This has now been mentioned in the manuscript.

Reviewer’s comment: There are a lot of comparisons performed in this investigation, so setting your alpha at .05 really runs the risk of increasing your family-wise error rate. The authors should not feel pressured to "find significance" in a pilot study. Again, the main purpose will be to provide effect sizes to be used in an a priori analysis for the NEXT study and to make improvements on the methodology going forward. I would rather see you be more conservative (e.g., Bonferroni correction per analysis).

Author’s response: This is a good suggestion, a greater emphasis has now been placed on effect sizes rather than statistical inferences (e.g., comparison of p-value to an a priori set alpha level) to improve the interpretation of our results in the Discussion section. With respect to Bonferroni correction, we only compared measures between groups at one time point and thus post-hoc tests did not apply in our analyses. 

RESULTS

Reviewer’s comment: Please provide another table with descriptive statistics of the participants and perform statistical comparisons between the groups (age, height, weight, etc.).

Author’s response: Thank you for highlighting this omission. The table has now been included in the manuscript. 

DISCUSSION

The discussion is quite lengthy, but well-written. If the authors are struggling with a word count while trying to address any comments, I would recommend cutting this down a bit.

Reviewer’s comment: Please add a Limitations section prior to your Conclusion.

Author’s response: Limitations have now been added to the manuscript. 

Reviewer’s comment: Reframe your Discussion and Conclusions per the pilot nature of this study. This is probably not a definitive investigation on this topic. Temper your conclusions appropriately. See notes above.

Author’s response: Given that this is a pilot study the hypothesis has been removed and the emphasis within the discussion and conclusion is now on the aim. 

Reviewer #2: This submission examines the gaze behavior during ambulation tasks of persons with osteoarthritis of the knee and compares these behaviors (as well as some gait metrics) with age matched controls without OA of the knee. 

Reviewer’s comment: The submission shares some interesting observations but it suffers from the lack of a central theory and in its current state, does not seem to present or test a clear hypothesis.

Author’s response: Given that this is a pilot study the hypothesis has been removed and the emphasis within the discussion and conclusion is now on the aim. 

Reviewer’s comment: The introduction simply states that people with OA present with weakness, pain, abnormal gait and are at an increased risk for falls. It follows with a discussion of the presence of differences in gaze behaviors and balance when comparing older adults at risk for falls and younger people and finishes with a proposal that gaze behaviors in persons with knee OA have not been studied. There is no discussion of a proposed relationship between living with knee OA and the development of maladaptive gaze behaviors. In the absence of a literature on such a specific question, presenting a broader discussion of CNS adaptations to chronic orthopedic conditions could help a reader make the connection between the two phenomena being discussed. Without this grounding in theory it is difficult to see this submission as something beyond a report of two co-occurring conditions in older persons.

Author’s response: Thank you for highlighting this omission. The introduction how includes CNS adaptations to OA with a proposed link to gaze. 

Reviewer’s comment: This submission presents dozens of statistical comparisons. The authors should consider focusing their paper to test a primary hypothesis and identify some key secondary hypotheses and analyses. Sticking with the current shot gun approach would require presenting corrected p-values.

Author’s response: This is a good suggestion, a greater emphasis has now been placed on effect sizes rather than statistical inferences (e.g., comparison of p-value to an a priori set alpha level) to improve the interpretation of our results in the Discussion section. With respect to Bonferroni correction, we only compared measures between groups at one time point and thus post-hoc tests did not apply in our analyses. 

Reviewer’s comment: The description of the two samples needs to be more extensive. Critical information such as subjects and controls comorbidities (particularly DM), fall history, and the presence or absence of other orthopedic issues needs to be shared, as does some measure of the severity of knee OA in the OA group and the length of time they have suffered from OA.

Author’s response: This is a good suggestion for us to include in the full study. With regards to this pilot study, we have matched age, sex, heigh and weight between the groups which we believe will be a sufficient base for our full study. 

Reviewer’s comment: More information is needed on the system utilized to collect gaze behaviors. How does the equipment work? Is there published data speaking to the reliability and validity of measurements collected during gait, with the system, in the studied population?

Author’s response: We have included reliability of the system to study gaze behaviors with references which includes the equipment and more information. 

Reviewer’s comment: The analyses of dual task data might make more sense if the authors analyzed the differences in dual task cost as opposed to citing abnormal single task data, abnormal dual task data and proposing a possible central neural difference causing the abnormal dual task data.

Author’s response: This is a good suggestion, as a pilot study there are practical restrictions however we can certainly consider this is in the full study. 

Reviewer’s comment: The FRT analysis does not fit into this paper. A static limit of stability test with the subject in a wide stance does little to inform a discussion of stability in gait, a much higher order of balance. I could make a similar argument against keeping the MVC analyses. People with KOA present with decreased force generation abilities. 

Author’s response: Thank you for this suggestion. FRT has been utilized in the literature to discriminate between fallers and non-fallers which we believe may have been useful in our future study. This suggestion will be taken on, especially with regards to future research. 

6. PLOS authors have the option to publish the peer review history of their article (what does this mean?). If published, this will include your full peer review and any attached files.

Do you want your identity to be public for this peer review? For information about this choice, including consent withdrawal, please see our Privacy Policy.

Reviewer #1: No

Reviewer #2: No

---

## [Decision Letter · Decision Letter 1]

26 Jan 2023

PONE-D-22-25749R1Patients with Knee Osteoarthritis have altered Gait and Gaze Patterns compared to Age-Matched Controls: A Pilot Study.PLOS ONE

Dear Dr. Le Rossignol,

Thank you for submitting your manuscript to PLOS ONE. After careful consideration, we feel that it has merit but does not fully meet PLOS ONE’s publication criteria as it currently stands. Therefore, we invite you to submit a revised version of the manuscript that addresses the points raised during the review process.

We look forward to receiving your revised manuscript.

Kind regards,

Ravi Shankar Yerragonda Reddy, Ph.D

Academic Editor

PLOS ONE

Additional Editor Comments:

The second reviewer has recommended rejection since the authors have not addressed his comments. An further opportunity to respond to the comments is provided. Please review the suggestions and respond accordingly.

Reviewers' comments:

Reviewer's Responses to Questions

**Comments to the Author**

1. If the authors have adequately addressed your comments raised in a previous round of review and you feel that this manuscript is now acceptable for publication, you may indicate that here to bypass the “Comments to the Author” section, enter your conflict of interest statement in the “Confidential to Editor” section, and submit your "Accept" recommendation.

Reviewer #1: All comments have been addressed

Reviewer #2: (No Response)

2. Is the manuscript technically sound, and do the data support the conclusions?

Reviewer #1: (No Response)

Reviewer #2: Partly

3. Has the statistical analysis been performed appropriately and rigorously? 

Reviewer #1: (No Response)

Reviewer #2: Yes

4. Have the authors made all data underlying the findings in their manuscript fully available?

Reviewer #1: (No Response)

Reviewer #2: Yes

5. Is the manuscript presented in an intelligible fashion and written in standard English?

Reviewer #1: (No Response)

Reviewer #2: Yes

6. Review Comments to the Author

Reviewer #1: (No Response)

Reviewer #2: Cursory attempts to address half of my suggestions have been made. A theoretical underpinning for this study is still lacking. A coherent examination of a theory is still lacking. A rigorous description of the subjects that supports the possibility that a central nervous system adaptation to knee pain might have occurred is still lacking. A rigorous examination of the impact of dual task conditions is still lacking.

7. PLOS authors have the option to publish the peer review history of their article (what does this mean?). If published, this will include your full peer review and any attached files.

Reviewer #1: No

Reviewer #2: No

<quillbot-extension-portal></quillbot-extension-portal>

---

## [Author Response · Author response to Decision Letter 1]

26 Feb 2023

We agree with the comments and suggestions from reviewer #2; addressing these comments and suggestions have strengthened the support for central theory within our research. Accordingly, we have expanded our introduction to address this concept with further emphasis on the integration with the central nervous system and duals tasks in in relation to KOA. 

Thank you for your assistance with improving the central theory.

---

## [Decision Letter · Decision Letter 2]

8 Mar 2023

Patients with Knee Osteoarthritis have altered Gait and Gaze Patterns compared to Age-Matched Controls: A Pilot Study.

PONE-D-22-25749R2

Dear Dr. Scott Le Rossignol,

We’re pleased to inform you that your manuscript has been judged scientifically suitable for publication and will be formally accepted for publication once it meets all outstanding technical requirements.

Kind regards,

Ravi Shankar Yerragonda Reddy, Ph.D

Academic Editor

PLOS ONE

Additional Editor Comments (optional):

The authors have addressed all the comments raised by the reviewers, and the quality of the manuscript has improved. The manuscript is accepted in its current form.

Reviewers' comments:

Reviewer's Responses to Questions

**Comments to the Author**

1. If the authors have adequately addressed your comments raised in a previous round of review and you feel that this manuscript is now acceptable for publication, you may indicate that here to bypass the “Comments to the Author” section, enter your conflict of interest statement in the “Confidential to Editor” section, and submit your "Accept" recommendation.

Reviewer #1: All comments have been addressed

Reviewer #2: All comments have been addressed

2. Is the manuscript technically sound, and do the data support the conclusions?

Reviewer #1: (No Response)

Reviewer #2: (No Response)

3. Has the statistical analysis been performed appropriately and rigorously? 

Reviewer #1: (No Response)

Reviewer #2: (No Response)

4. Have the authors made all data underlying the findings in their manuscript fully available?

Reviewer #1: (No Response)

Reviewer #2: (No Response)

5. Is the manuscript presented in an intelligible fashion and written in standard English?

Reviewer #1: (No Response)

Reviewer #2: (No Response)

6. Review Comments to the Author

Reviewer #1: (No Response)

Reviewer #2: (No Response)

7. PLOS authors have the option to publish the peer review history of their article (what does this mean?). If published, this will include your full peer review and any attached files.

Reviewer #1: No

Reviewer #2: No

<quillbot-extension-portal></quillbot-extension-portal>

---

## [Editor Report · Acceptance letter]

13 Apr 2023

PONE-D-22-25749R2 

Patients with Knee Osteoarthritis have altered Gait and Gaze Patterns compared to Age-Matched Controls: A Pilot Study. 

Dear Dr. Le Rossignol:

I'm pleased to inform you that your manuscript has been deemed suitable for publication in PLOS ONE. Congratulations! Your manuscript is now with our production department. 

Kind regards, 

on behalf of

Dr. Ravi Shankar Yerragonda Reddy 

Academic Editor

PLOS ONE